# swAFL: A Library of High-Performance Activation Function for the Sunway Architecture

Jinchen Xu [1], Fei Li [1], Ming Hou [1] and Panjie Wang [2,*]

[1] State Key Laboratory of Mathematical Engineering and Advanced Computing, Zhengzhou 450002, China
[2] Zhengzhou Xinda Institute of Advanced Technology, Zhengzhou 450001, China
* Correspondence: wangpanjie1995@163.com

**Abstract:** The Sunway supercomputers have recently attracted considerable attention to execute neural networks. Meanwhile, activation functions help extend the applicability of neural networks to nonlinear models by introducing nonlinear factors. Despite the numerous activation function-supported AI frameworks, only PyTorch and TensorFlow were ported to the Sunway platforms. Although these libraries can meet the minimum functional requirements to deploy a neural network on the Sunway machines, there still exist some drawbacks including the limited number of usable functions and unsatisfactory performances remaining unresolved. Therefore, two activation function algorithms with different computing accuracies were developed in this study, and an efficient implementation scheme was designed using the single instruction/multiple data extension and multiply–add instructions of the platform. Finally, an efficient library-swAFL-composed of 48 function interfaces was designed and implemented on the Sunway platforms. Experimental results indicate that swAFL outperformed PyTorch and TensorFlow by 19.5 and 23 times, respectively, on average.

**Keywords:** Sunway platform; activation function; interval transformation; interval partitioning





## 1. Introduction

Activation functions [1] introduce nonlinear factors into neural networks to expand the scope of their application. With the rapid development of AI applications, several activation function-supported AI frameworks have been invited and well developed, such as PyTorch [2], TensorFlow [3], Caffe [4], Microsoft CNTK [5], Theano [6], and Keras [7]. Derived from Torch and launched in 2017 by the Facebook Institute of Artificial Intelligence, Pytorch wraps 21 activation functions in its torch.nn.functional module. TensorFlow is a symbolic mathematics system based on dataflow programming, widely used by the machine learning community to develop novel neural network algorithms; its tf.keras.activations [8] module also offers a wide class of activation functions. Caffe is an open-source deep learning framework written in the C++ language; its activation layer contains six commonly used activation functions. Keras is a neural network interface written in Python, and its Activations class provides 12 defined activation functions [9]. The number of such machine learning frameworks is also growing rapidly in the industry society: Huawei introduced a heterogeneous computing architecture, CANN, for various AI scenarios; Baidu launched PaddlePaddle, an open-source deep learning platform derived from industry practices. The commonness of these works is well-developed support for commonly used activation functions.

The next-generation machine of Sunway, TaihuLight, uses a totally new high-performance heterogeneous many-core processor called SW26010-Pro, whose chipset comprises six core groups (CG), each including one management processing element that controls the operations and 64 slave cores that perform computational tasks. The processor uses SW64 instruction sets and supports 256/512-bit single instruction/multiple data (SIMD) and multiply-add operations. Regarding floating-point numbers, SW26010-Pro processors support

double- and single-precision operations on both the master and slave cores, as well as half-precision operations on the slave cores [10].

As such, developing a library of high-performance activation functions on such machines is emerging for the integration of high-performance computing and artificial intelligence on the Sunway platforms. With the rise of Sunway TaihuLight and its new generation machine, the Sunway supercomputer family have been widely considered as a new high-performance computing platform, and their demand for more efficient activation function libraries also increases. Currently, the two mainstream AI frameworks, PyTorch and TensorFlow, have been ported onto the Sunway platform. However, compared with the Intel x86 platform, the inventory of ported activation functions on this platform faces two challenges: first, the number of functions is small, making the programming requirements not fully satisfied; second, the execution performance of the activation functions on the Sunway platform is often 19–50 times slower than on the Intel x86 platform, even with both the processors working at the same frequency. Both these challenges limit the implementation of AI applications on the Sunway platform.

In view of the above problems, this study designed a library of activation functions called swAFL, which can further enrich the software ecosystem of the Sunway platform. The contributions of this study include:

- A library of activation functions-swAFL-designed and implemented on the Sunway platform. It provided three types of precision configurations: single-precision operations on the master core, and both, with a total of 48 function interfaces.
- Both computing accuracy and function performance were targeted, and two efficient algorithms were proposed considering the different levels of demands: the interval transformation-polynomial approximation algorithm for single-precision activation functions, and the interval partition-polynomial approximation-look-up table algorithm for half-precision activation functions.
- By combining the SIMD extension and fast multiply-add (FMA) instructions of the Sunway platform, the activation functions could run efficiently. In terms of the average speedup ratios, it outperformed PyTorch on the same platform by 6.4, 15.2, and 37.6 times, and TensorFlow by 14.2, 32.1, and 24 times, for single-precision operations on the master core, and half- and single-precision operations on the slave cores, respectively.

## 2. Background

### 2.1. Half-Precision Floating Point

Recently, half-precision floating-point numbers have been widely used. They have two formats: the extended half-precision (FP16) from the IEEE 754 standard, and the brain floating point (BF16), which is specific for deep learning.

FP16 [11] was proposed by NVIDIA in 2002 to reduce data transmission and memory consumption. Its application scenarios generally do not require high computation accuracy. The BF16 half-precision floating-point format [12] was developed by Google Brain, which is an AI research group of Google. Even though higher levels of computing accuracy can be achieved with both single- and double-precision floating-point numbers, they require more expensive costs in terms of both time and space. In addition, such a high accuracy is usually not necessary in the field of deep learning and achieving a balance between a satisfactory accuracy and time and space costs can greatly improve both the learning and inference speeds. Consequently, BF16 was developed to provide an optimum solution. Because they are half-precision floating-point formats, both have fewer available bits: FP16 has more mantissa bits, whereas BF16 has a reduced number of mantissa bits but the same number of exponential bits as that of the single-precision format. Owing to these differences, BF16 has a larger range of numbers than FP16.

### 2.2. Unit in the Last Place (ULP)

Floating-point numbers cannot accurately represent all values because some operations, such as round-off and arithmetic, will result in certain errors. Therefore, errors are inevitable in floating-point calculations. ULP [13] has generally been used as an index to measure the error in floating-point calculations. Since the emergence of its concept, there have been several versions of its definition. The definition of ULP adopted in this study is as follows:

$$ulp(x) = \beta^{\max(e, e_{\min}) - p + 1}, |x| \in \left[ \beta^e, \beta^{e+1} \right) \tag{1}$$

where $\beta$ represents the cardinal, $e_{\min}$ represents the minimum exponent, and $p$ represents the precision.

As shown in Figure 1, $ulp(result) = |res\_right - res\_left|$. Therefore, the calculation error of the function is represented as $error = (res\_c - result)/(ulp(result))$.

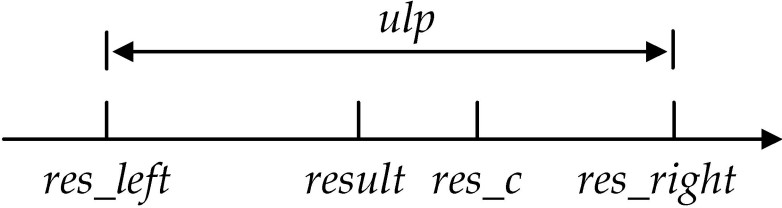

**Figure 1.** Illustration of ULP. *result* is the accurate calculation result of the function when the input is ln _*one*, whereas *res_left* and *res_right* denote the two floating-point numbers closest to the accurate result's left and right, respectively. *res_c* represents the approximated result when the input of the function is ln _*one*.

If the calculation is correct, the minimum error of the function is less than 0.5 ulp. Moreover, if the error of a function is always smaller than 0.5 ulp, the function can constantly return the floating-point number closest to the accurate result—in other words, the function is properly rounded.

### 2.3. Activation Function

An activation function is a function that acts on a neuron of a deep neural network. In the absence of an activation function, the output of each layer in a neural network is linearly related to the input of its previous layer. The activation function introduces a nonlinear factor into the neural network, which can implement the nonlinear mapping between the input and output of the neural network. This makes the neural network able to approach any non-linear function, and it is thus helpful to apply the neural network to more non-linear models. Currently there are various types of activation functions. This paper classifies the commonly used activation functions into two categories represented by Sigmoid and ReLU functions based on the connection between the function's own nature, the function implementation algorithm and the algorithm used to implement the swAFL activation function library function in this paper. For functions that have different expression characteristics from Sigmoid functions and ReLU functions, we classify them into other categories, such as VAF, APL, KAF, etc.

The Sigmoid activation-based functions include HardSigmoid, Swish, tanh, Softmax, and Softsign. Primarily used in binary classification problems, the Sigmoid function [14] maps the input value to the interval [0, 1]. This function has a smooth image and is easy to derive and is the closest function to biological neurons. Since the mean value of the Sigmoid function saturates the deep network, especially the topmost hidden layer, it is not suitable for deep networks with random initialization. The Softmax function [15], also called the normalized logic function, converts the original output into probabilities. Since its introduction, several improved versions have been proposed, including hierarchical Softmax (H-Softmax) [16] by Morin and Bengio inspired by the binary tree, Differentiated Softmax (D-Softmax) [17] with lower complexity by Chen et al. Softmax loss model [18]

based on character-level CNNs designed by Jozefowicz, and adaptive Softmax proposed by Grave et al. All the new functions were reported to have constantly improved the computational efficiency of the Softmax function. Additionally, the HardSigmoid function [19] has been introduced as the linear piecewise approximation of the Sigmoid function. The Tanh function [20] is an expanded version of the Sigmoid function, but the gradient vanishing problem is still not solved. However, the difference from Sigmoid function is that Tanh function' average value is 0 so it has better effect in practical application. The Tanh function is gradually replacing the Sigmoid function as the standard activation function in classification tasks. Considering the possible disappearance of the gradient of the above functions, Glorot X et al. [21] proposed the Softsign function that has smoother asymptotes. Thereafter, the Swish function was proposed as another variant of the Sigmoid function [22]. It is defined as *Swish (x) = x · Sigmoid (βx)*, where *β* is a constant or trainable parameter and the function image is smooth and differentiable everywhere. Swish functions have no upper bound or lower bound, and unlike the other common activation functions, they are non-monotonic.

The ReLU activation-based functions [23] include ReLU, LeakyReLU, Softplus, scaled exponential linear unit (SeLU), exponential linear unit (ELU), and ReLU6. Because Sigmoid and its similar functions must perform a large number of calculations in back propagations for the error gradient, and the gradient is likely to disappear: they can seldom accomplish the training of deep networks. Therefore, Glorot et al. [24] employed the ReLU function to solve the gradient disappearance problem. However, ReLU also has problems, such as neuron death and the mean of the output not being zero. For example, the output range is $[0, +\infty]$ in large network environment, only ReLU function is used, which will result in numerical explosion. In order to solve this problem, the ReLU6 function is put forward, the difference between the ReLU6 function and the ReLU function is that the maximum output of the limit function is 6. Therefore, there are constant efforts to improve ReLU. Banerjee et al. [25] proposed the polyphase ReLU activation function, Clevert et al. [26] presented the ELU function, and Klambauer et al. [27] introduced the SeLU function. The SeLU function is the activation function of the self-normalized neural network, the λ and α in the function expression are fixed values obtained by formula proof, not the values obtained by training. Additionally, the LeakyReLU function [28] has been developed to prevent the dying ReLU problem, and the Softplus function has been used to prevent inactivated neurons. The Softplus function is a smooth version of the ReLU function that returns any value greater than 0. Unlike the ReLU function, Softplus is differentiable everywhere, and its derivatives are continuous and non-zero, thus preventing the emergence of silent neurons. However, because the Softplus function is asymmetric, non-zero-centered, and the derivative is often less than 1, it may lead to gradient disappearance problems.

In addition to these, there are also some other activation functions such as PReLU [29], RReLU [30], CReLU [31], Mish [32], and DY-ReLU [33].

## 3. Algorithm Design for Activation Functions

Activation functions can be divided into two algorithm design categories. The first is related to transcendental functions, such as exp and tanh, including the ELU, Softplus, Sigmoid, LogSigmoid, SeLU, Softmax, Ttanh, Mmish, and Swish functions. The second category comprises numerical functions that are independent of transcendental functions; these include the HardSigmoid, ReLU, ReLU6, LeakyReLU, Softsign, Hardshrink, Softshrink, ThresholdedReLU, and PReLU functions. Most of these are numerical processing functions. Through the judgment and classification of the input value, implementation of the algorithm is intuitive and simple.

Because the algorithms used in the first category of activation functions are closely related to computation accuracy and performance, the algorithm design of the single- and half-precision functions of this category were discussed in this study.

For single precision input, the calculation interval is transformed from the definition domain to the approximated interval according to the property of the function, and the

value of the function is calculated by using the min-max error approximate polynomial for the data in the approximated interval, then the function values in the calculated interval are obtained by combining the function properties. The input value x ∈ D is transformed into x\* ∈ D$_P$, and the data in the approximated interval are approximated by a polynomial. The implementation of the "interval transformation-polynomial approximation is as Algorithm 1:

---

**Algorithm 1.** Interval transformation-polynomial approximation

---

Input: $x$
Output: $f(x)$

1.    x ∈ D → $x^* ∈ D_p$

// Converts the calculated interval of an input value from the domain D to an approximate interval D$_p$

2.    POLY$(x^*) → f(x^*)$

// The polynomial approximation algorithm is used to calculate the function value for the input value in the approximate interval

3.    $f(x^*) → f$(x)

// The function values in the interval are obtained by using the function properties and the function values in the approximate interval

---

For the half-precision input, the method of interval division and look-up table is adopted in the whole. The interval division technique divides the computing interval into an approximated interval and multiple look-up table intervals according to the numerical law of floating-point numbers, x ∈ D is converted to x\* ∈ D$_P$. After the interval operation is completed, the data in the approximate interval are calculated by polynomial approximation algorithm, and the function values in the computing interval are obtained by mapping, and the function values in the table-looking interval are obtained by table-looking algorithm. "Interval partition-polynomial approximation-look-up table" is implemented as Algorithm 2:

---

**Algorithm 2.** Interval partition-polynomial approximation-look-up table

---

Input: x
Output: f(x)

1.    x ∈ D → $x^* ∈ D_p, x_1 ∈ D_1, x_2 ∈ D_2 \ldots , x_n ∈ D_n$

// The computing interval is divided into approximate interval and look-up interval

2.    POLY$(x^*) → f(x^*)$

// The polynomial approximation scheme is used in the approximate interval

3.    lookup_table$(x_i) → f(x_i)$, i = 1, 2, \ldots , n

// Use the lookup algorithm within the lookup interval

4.    $f(x^*), f(x_1), f(x_2), \ldots , f(x_n) →$ f(x)

// Use the function properties to get the value of the function within the calculated interval

---

In the above algorithm, POLY() represents a polynomial approximation function, and lookup_table() represents a lookup table function.

### 3.1. Algorithms for the Single-Precision Activation Function

For single-precision input, the calculation interval is first transformed from the defined interval [a, b] to the approximate interval [a', b'], based on the function's properties. Next, the approximate polynomial is calculated using Sollya. Finally, the function values in the

approximate interval are reconstructed based on the properties of the function. The details of the algorithm are shown in Figure 2.

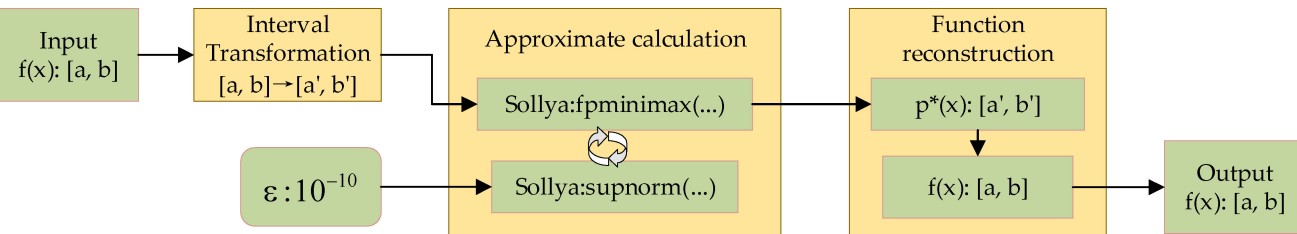

**Figure 2.** Implementation Framework of Single-Precision Activation Function Algorithm.

### 3.1.1. Interval Transformation

The purpose of interval transformation is to reduce the calculation interval from the domain of the function to a limited interval. The idea is to make use of the mathematical properties of the function and complete it through transformations. In practice, it can be divided into additive and multiplicative reductions [34].

Additive reduction is expressed by $x^* = x - kC$, where $k$ is an integer, and $C$ is a constant. When applied to trigonometric functions, $C$ is a multiple of $\frac{\pi}{4}$. For example, the domain of $\sin(x)$ can be reduced from $x \in (-\infty, +\infty)$ to $x^* \in \left[0, \frac{\pi}{4}\right]$.

In contrast, multiplicative reduction is expressed by $x^* = \frac{x}{kC}$, where $k$ is an integer, and $C$ is a constant. When applied to logarithmic functions, $C$ is the cardinality. For example, $\exp(x)$ can be reduced from $x \in (-\infty, +\infty)$ to $x^* \in \left[-\frac{\ln 2}{2}, +\frac{\ln 2}{2}\right]$.

The basic idea of interval transformation is to utilize the mathematical properties of the function. Considering the exponential function as an example, its properties include: $e^{a+b} = e^a \times e^b$ and $e^{mn} = (e^m)^n$. Based on the properties of the function, the input value can be split to construct $e^{\ln 2}$ for transformation through the following steps:

The domain of the function $exp$ is $(+\infty, -\infty)$, and the input $x$ is split into two parts: $num$ and $remainder$, where $num = \left|\frac{x}{\ln 2}\right|$, and $remainder = x - num * \ln 2$. If $x = num * \ln 2 + remainder$, the function can be expressed as follows:

$$\exp(x) = \exp(num * \ln 2 + remainder). \tag{2}$$

The calculation interval of the input x is $(-\infty, +\infty)$. Utilizing the property $\exp(x + y) = \exp(x) \times \exp(y)$, Equation (2) can be transformed as follows:

$$\exp(x) = \exp(num \times \ln 2) \times \exp(remainder), \tag{3}$$

which converts the computation of x into the product of num $\times$ ln2 and the remainder. Because $\exp(\ln 2) = 2$, Equation (3) can be further transformed as follows:

$$\exp(x) = 2^{num} * \exp(remainder). \tag{4}$$

Following these steps, the calculation interval of the function exp is transformed from $(-\infty, +\infty)$ to $[-\ln 2/2, \ln 2/2]$, where $2^{num}$ can be calculated at a high speed through bitwise shift, and $\exp(remainder)$ can be calculated through polynomial approximation.

### 3.1.2. Polynomial Approximation

After obtaining the approximate interval $D_p$ through the interval transformation steps introduced in the previous section, the approximation of the continuous function f(x) is calculated by the polynomial $p^*(x)$ at the approximate interval. The "distance" between the polynomial approximation and the function value, i.e., the difference between the two values, is represented as $D = ||f(x) - p^*(x)||$. If this difference is at its minimum, $p^*(x)$ is regarded as the optimum polynomial approximation scheme, which was adopted

in this study. The commonly used polynomial approximation schemes are Taylor, minimax error [35], and Chebyshev [36]. The Taylor polynomial approximation algorithm approximately calculates the function value through its Taylor expansion. However, its convergence speed is relatively slow, which makes it difficult to achieve high precision in low-order expansion polynomials. Therefore, it is not suitable for applications that require higher precision and speed. The average error obtained by the Chebyshev polynomials is the smallest, and the maximum error obtained by minimax error polynomials is the minimum. The activation function designed and implemented in this study aimed to achieve the minimum maximum error; therefore, the minimax error polynomial was selected as the approximate calculation scheme.

Sollya [37] is a mature interactive tool that can handle numerical functions with arbitrary precision. It can accurately evaluate functions, realize various polynomial approximation schemes of mathematical functions and expressions, and obtain the absolute error or relative error of the scheme, automatic implementation of the mathematical library for the polynomial and drawing functions. The Sollya tool provides multiple precision versions, including half-precision, single-precision, double-precision, and extended double-precision, for basic functions as well as mathematical expressions. In this study, the minimax error polynomials were generated using Sollya, and the relative errors of the generated polynomial schemes were evaluated using the generated polynomials. Through an iterative calculation process, and combined with the range of single-precision relative error (which should be controlled within $10^{-10}$ [38]), the approximate polynomials that meet the requirements of error calculation were generated. Considering the exp function's calculation as an example, the process to generate the minmax polynomial is shown in Figure 3.

```
1.>P=fpminimax(exp(x),8,[|SG...|],[-log(2)/2; log(2)/2]);
2.> supnorm(p,exp(x), [-log(2)/2; log(2)/2],relative,2^(-40));
3.>[9.6738433931443193174030602027565e-12;9.6738433931443193174030602e-12]
4.> printexpansion(P);
5.>0x3ff0000000000000+x*(0x3ff0000000000000+x*(0x3fe0000000000000+x*(0x3fc5555560000000+x*
(0x3fa5555580000000+x*(0x3f8110fec0000000+x*(0x3f56c0f000000000+x*(0x3f2a278920000000+x*0
x3efab38ba0000000)))))))
```

**Figure 3.** The min-max polynomial generation process of the exp function.

Figure 3 illustrates the process of using Sollya to find the minimax error polynomials and the errors of the exp function in the interval $[-\ln 2/2, \ln 2/2]$. The first line obtains the 8th order minimax error polynomial of the exp function in the defined interval, wherein SG refers to single precision, and rounds its parameters to the closest number. The second line evaluates the relative error of the polynomial. The fourth and fifth lines output the resulting polynomial.

Specifically, polynomial generation is realized by calling the function fpminimax (f, n, formats, range). It calculates the approximate polynomial of the function in the interval range, and the largest order is n. "formats" is used to specify the floating-point data type of the polynomial coefficients; for example, D refers to double precision. After generating a polynomial of the specified precision and order, the supnorm function is used to evaluate the relative error of the approximate polynomial. The calculation error of the approximate polynomial usually decreases when order increases. Therefore, the smallest order of polynomial that meets error requirements can be determined as the final approximate polynomial.

### 3.2. Algorithms for the Half-Precision Activation Function

For half-precision floating-point numbers, based on their numerical characteristics, the calculation interval is divided into an approximate interval and several small lookup table (LUT) intervals. The calculation in the approximate interval uses the same algorithm

as the single-precision activation function, whereas the calculation of LUT interval by the following steps:

- Use the high-precision library to calculate the exact value of the function
- Store the exact value into the table
- Calculate the offset to obtain the data in the table.

### 3.2.1. Interval Partitioning

The half-precision floating-point numbers have the natural advantage of lower bit width. Considering FP16 as an example, excluding special values (NaN and inf), there exist 61,442 floating-point numbers, and the distribution of numbers exhibits certain rules: if the numbers with steps equal to their adjacent floating-point numbers are grouped, the count of floating-point numbers in each group is 1024. Then the difference between two adjacent numbers in one group is equal, and the difference between the next two adjacent numbers in the next group is 1/2 of the difference between the last two adjacent numbers, as shown in Table 1.

**Table 1.** The rule of half-precision floating-point numbers.

| Interval | The Number of Floating-Point Numbers in the Interval | Interval Between Two Adjacent Numbers |
| --- | --- | --- |
| $\left[-2^{15}-2^5, -2^{15}\right]$ | 1024 | $2^5$ |
| $\left(-2^{-15}, -2^{-14}\right]$ | 1024 | $2^4$ |
| $\left(-2^{-14}, -2^{-13}\right]$ | 1024 | $2^3$ |
| $\left(-2^{-13}, -2^{-12}\right]$ | 1024 | $2^2$ |
| $\left(-2^{-12}, -2^{-11}\right]$ | 1024 | $2^1$ |
| … … | | |
| $\left(-2^1, -2^0\right]$ | 1024 | $2^{-10}$ |
| $\left(-2^0, -2^{-1}\right]$ | 1024 | $2^{-11}$ |
| $\left(-2^{-1}, -2^{-2}\right]$ | 1024 | $2^{-12}$ |
| … … | | |

A total of 47% of half-precision floating-point numbers fall in the $(-1, 1)$ range, so it is not practical to use lookup table method for such dense data. Therefore, for half-precision FP16, if the calculation interval contains $(-1, 1)$, then $(-1, 1)$ should be set to an approximate interval. For the division of look-up table interval, the concrete method of look-up table is to calculate the offset of the input value relative to the starting point of the interval, according to the offset, get the position of the input value corresponding to the result in the table, and then obtain the function value. If the calculation of offset has too many branches, it will increase the performance overhead, so we should try to avoid such a scenario when using look-up table method. If all table lookup operations within an interval are the same offset calculation method, the number of branches will be reduced accordingly. Therefore, to avoid excessive branching, we need to start with the law of floating-point numbers and divide the data into a small interval according to the principle of calculating the expression with the same offset, that is, each row in the Table 1 can be divided into a small lookup interval $D_i$, which divides the data in the domain except the approximate interval into several sub-intervals.

The interval partitioning algorithm primarily distinguishes the following situations: (i) if all the input values are smaller than $-1$ or greater than 1, data of the same group are assigned to the same interval according to the half-precision floating-point rule, which directly forms the LUT interval. (ii) If $(-1, 1)$ is a subset of the input interval, then $(-1, 1)$ is set to an approximate interval, and the remaining input interval is divided into groups according to the above-mentioned half-precision floating-point rule, as stated in the first situation. (iii) If the input interval partially belongs to $(-1, 1)$, the input interval within $(-1, 1)$ is set to an approximate interval, and the rest of the interval is divided into groups according to the half-precision floating-point rule to form an LUT interval. (iv) If the input

interval is entirely covered by (−1, 1), it is only set to an approximate interval. The specific interval partitioning algorithm is as Algorithm 3:

---

**Algorithm 3.** Interval partitioning algorithm

---

Input: calculation interval (a, b)
Output: approximate interval $D_p$, LUT interval $D_i$

1.　　a_exp ← $[\log_2 |a|]$

2.　　b_exp ← $[\log_2 |b|]$

3.　　if (a < b && b < −1):

4.　　n ← a_exp − b_exp + 1

5.　　$D_1 \leftarrow (a, -2^{a\_exp-1})$;

6.　　for i=2 to n−1:

7.　　$D_i \leftarrow (-2^{a\_exp-i+1}, -2^{a\_exp-i}]$

8.　　$D_n \leftarrow (-2^{b\_exp+1}, b]$

9.　　$D_p \leftarrow$ NULL

10.　if(1<a&&a<b):

11.　n ← b_exp − a_exp + 1

12.　$D_1 \leftarrow (a, 2^{a\_exp})$;

13.　for i=2 to n−1:

14.　$D_i \leftarrow (2^{a\_exp-i-2}, -2^{a\_exp+i-1}]$

15.　$D_n \leftarrow (2^{b\_exp-1}, b]$

16.　$D_p \leftarrow$ NULL

17.　if(a<−1&&−1<b&&b<1):

18.　n ← a_exp

19.　$D_1 \leftarrow (a, -2^{a\_exp-1})$;

20.　for i=2 to n:

21.　$D_i \leftarrow (-2^{a\_exp-i+1}, -2^{a\_exp-i}]$

22.　　　$D_p \leftarrow (-1, b]$

23.　if(a<−1&&1<b):

24.　n ← a_exp + b_exp

25.　　　$D_1 \leftarrow (a, -2^{a\_exp-1})$;

26.　for i=2 to a_exp:

27.　$D_i \leftarrow (-2^{a\_exp-i+1}, -2^{a\_exp-i}]$

28.　for j=1 to n−1:

29.　$D_{j+a\_exp} \leftarrow (-2^{j-1}, -2^j]$

30.　$D_n \leftarrow (2^{b\_exp-1}, b]$

---

31.  $D_p \leftarrow (-1,1)$

32.  if($-1 < a$&&$a < b$&&$b < 1$):

33.      $D_p \leftarrow (a,b)$

34.  if($-1 < a$&&$a < 1$&&$1 < b$):

35.  n $\leftarrow$ b_exp

36.  for i=1 to n$-1$:

37.  $D_i \leftarrow (2^{i-1}, 2^i]$

38.  $D_n \leftarrow (2^{b\_exp-1}, b]$

Here, get_exponent () is a function to obtain the exponential bit of the input floating-point number, and the set_fraction(x) function sets the exponential bit of the input floating-point number to x and the mantissa to 0. D_i (i = 1, 2, ... , n) denotes multiple lookup intervals, and D_p represents approximate intervals.

### 3.2.2. Lookup Table

The table-lookup method is to pre-store the exact result of the input value to be calculated by the function into the table, and then look up the function result corresponding to the input value through the index. The precision of the result in the storage table and the stored value can be selected according to the strategy given by the user. When the value of the function is small, the function value can be obtained directly by calculating the offset of the input value relative to the starting point of the calculation interval. However, the memory space required to store the table has an exponential relationship with the data bit width of the input value. When the value range of the function is large, it is not practical to store all the input results in the table. Therefore, the interpolation method within the calculation interval is generally adopted: the function result of the interpolation point is stored in the table, and then the function value is obtained by looking up the table and fitting through the interpolation strategy. The denser the interpolation, the higher the computational precision. The following two points should be noted in the table lookup method:

- Establish a suitable data table, choose the size of table according to the situation of memory space resource;
- Find the trade-off between functions' storage space and time cost.

With an increase in LUT intervals, the corresponding judgment branches will also increase linearly. The focus is to determine both the LUT interval that the input value belongs to and the offset within the interval. To obtain the best table look-up performance, direct positioning is based on the distribution characteristics of half-precision floating-point numbers. For the input, the exponent of the input number is first obtained, from which both the step between adjacent numbers in the interval and the interval can be obtained. Thereafter, the offset can be calculated as follows:

$$offset = (input + 2^{11-log_2 step} - step)/step \tag{5}$$

Considering the interval $[-4, -1]$ as an example, the step of two adjacent numbers within $[-4, -2]$ is $2^{-9}$, whereas that within $[-2, -1]$ is $2^{-10}$. If the input value input belongs to $[-4, -2]$, the offset value is calculated as $(input + 2^2 - 2^{-9})/2^{-9}$; and if the input value falls within the range of $[-2, -1]$, the offset value is $(input + 2^1 - 2^{-10})/2^{-10}$.

Consider the ELU function as an example. Compared with the direct LUT calculation, the aforementioned non-branch LUT algorithm can provide an effective acceleration for four different LUT intervals, as presented in Table 2. The test set was composed of

10,000 randomly generated half-precision floating-point numbers within the calculation interval, and the number of ticks for function execution was used as the measurement index.

**Table 2.** Performance comparison of non-branch and traditional LUT for ELU function.

| Interval | Ticks | | Speedup Ratio |
|---|---|---|---|
| | **Traditional LUT** | **Non-Branch LUT** | |
| $(-17, 32)$ | 47 | 37 | 1.27 |
| $(-16, 16)$ | 61 | 47 | 1.30 |
| $(-16, 32)$ | 46 | 36 | 1.28 |
| $(-8, 16)$ | 52 | 42 | 1.24 |

## 4. Implementation of Activation Functions on the Sunway Platform

The efficient calculation of an activation function relies on the support of SIMD extension instructions. During implementation, a thorough utilization of the SIMD extension components on the Sunway platform is essential for high-performance computing of functions. The specific implementation framework was divided into two parts: input processing and function implementation. The general framework of the algorithm is as Algorithm 4:

---

**Algorithm 4.**

---

Input: input
Output: output

1. /* Obtain the input array length M, number of the redundant items flag, N is the vector length of the SIMD instruction*/

2. M ← getLen(input)

3. flag = M%N

4. /*Call SIMD activation function in a loop*/

5. **for**(i = 0; i < =M -N -1; i+N) {

6. simd_load(vin, &(input[i]))

7. output ← SIMD_FUN(vin)

8. simd_store(vout, &(out[i]))

9. }

10. /* Call the scalar activation function on elements with insufficient vector length*/

11. **if** (flag != 0)

12. **for** (i = M-flag; i < M; i++)

13. out[i] ← basic_fun(input[i])

---

The first step is to complete the input processing. For the SIMD extension unit, N represents the data that can be simultaneous processed by an SIMD instruction, M represents the length of the input data, and n is M-M%N, which indicates the number of complete vectorizations in the input data. "flag" = M%N, which indicates the part that cannot be fully vectorized, i.e., the redundant items. If the number of scalar inputs is flag, the number of times to call the scalar function is also flag. The next step is to complete the function calculation, which consists of two parts: the cyclic calculation of the SIMD version and the calculation of the scalar version. For the part that can be completely vectorized, the SIMD activation function is called; whereas for the part that cannot be completely vectorized, the scalar function is called. Although the algorithms of the SIMD and scalar activation functions share the same concept, the instructions used in their implementations

are different. To illustrate the implementation of activation functions, the SIMD activation function is presented as an example.

### 4.1. Multi-Type Calculation Interval Mapping

Interval mapping includes not only the interval transformation of single-precision floating-point numbers but also the interval partitioning of half-precision floating-point numbers. For single-precision floating point numbers, it is necessary to map the calculation interval from the definition domain to the approximate calculation interval. The basic idea of interval transformation is to use the mathematical properties of the function to transform it, which primarily involves multiplication, addition, and division. Combined with the FMA instructions supported by the Sunway platform, division can be further converted into multiplicative operations. Considering the exp function as an example, the interval transformation separates the input x into two parts: num and remainder, where $num = [x/ln2]$ and $remainder = x - num \times ln2$. The detailed implementation is as Algorithm 5:

---

**Algorithm 5.**

---

1. Input: x
2. Output: num, remainder
3. num = simd_vams(x,R_LN2,12582912);
4. vparm2 = simd_vams (vparm0,(-L2U),x);
5. Remainder = simd_vams (vparm0,(-L2L),vparm2);

---

Where simd_vams(x,y,z) denotes the operation $x \times y + z$ and R_LN2 represents $1/ln2$. In the multiply-add instruction, 12582912 is added to convert the result to an integer.

For half-precision floating-point numbers, the calculation interval is mapped from the definition domain to an approximate interval and multiple LUT intervals through interval partitioning. Considering the ELU function on the interval $(-4, -1)$ as an example, the input from the positive interval is not tweaked, and the result of the input from the negative interval is set as $\alpha(e^x - 1)$. According to the algorithm introduced in Section 3.2.1, the interval can be directly assigned as the LUT interval, and based on the rule of half-precision floating-point numbers, it can be further divided into $(-4, -2)$ and $(-2, -1)$, as presented in Table 3.

**Table 3.** ELU function to calculate the interval of $(-4, -1)$.

| Interval | Reduced Intervals | Exponential Bits | Count of Numbers | Step between Adjacent Numbers |
|---|---|---|---|---|
| $(-4, -1)$ | $(-4, -2)$ | 10001 | 1024 | $2^{-9}$ |
| | $(-2, -1)$ | 10000 | 1024 | $2^{-10}$ |

### 4.2. Approximate Calculation of Polynomials

The commonly used polynomials are Taylor series, Chebyshev polynomials and minimax error polynomials. The Taylor series coefficients have a simple form and are easy to calculate. The approximate error of the Taylor series near the initial point is small and increases as it moves away from the initial point. The Chebyshev polynomial is an approximate polynomial with the smallest mean error. Minimax polynomials are approximate polynomials with the smallest max-error.

To ensure the correctness of floating-point calculations, our approach is more concerned with the maximum error of the function, rather than the average error. Therefore, this study adopts the minimum maximum error polynomial as the basis of polynomial approximation calculation. The minimax polynomials are defined as follows.

Take $\omega(x) = 1$, have $||f - p||_\infty = \max\limits_{x \in [a,b]} |f(x) - p(x)|$, the goal is $||f - p^*||_\infty = \min\limits_{p \in P} ||f - p^*||_\infty$. If such a polynomial $p^*$ exists, then it is called the nth order minimum maximum error polynomial of *f(x)* on the interval [a, b].

It is important to note that the coefficients of minimax polynomials are mostly inaccurate in floating-point format and have representation errors. If the rounding operation is not performed properly, this error will gradually spread and accumulate with the polynomial operation, which in turn will lead to a decrease in the accuracy of the final computation of the polynomial. Brisebarre proposed a new method to improve the floating-point implementation of minimax polynomials [39], which effectively avoids the precision loss caused by improper rounding of minimax polynomial coefficients in floating-point format, and integrated it into the fpminimax command of Sollya.

The approximate polynomial coefficients are generated using Sollya's fpminimax command. The input parameters of fpminimax include the target function name f, the approximation interval I and other parameters, and the output is the approximation polynomial for the function f in the interval I.

There are several different implementations of approximate polynomial calculation. By fully taking into account the properties of the Sunway platforms, our method implements the efficient polynomial calculation through the combination of the Horner [34] and Estrin [40] schemes. During the implementation, the FMA instruction supported by the platform was utilized such that the operation $x \times y + z$ can be realized with only one instruction. Moreover, only one rounding is required instead of two, which improves the accuracy. Construction of all levels of polynomials is as Algorithm 6:

---

**Algorithm 6.**

---

Input: $a_7$, $a_6$, $a_5$, $a_4$, $a_3$, $a_2$, $a_1$, $a_0$, x
Output: f(x) = $a_7 x^7 + a_6 x^6 + \cdots + a_1 x + a_0$

1.  #define MLA mla

2.  #define mla(x,y,z) simd_vmad(x,y,z)

3.  #define C2V(x) x

4.  #define POLY2(x, c1, c0) MLA(x, C2V(c1), C2V(c0))

5.  #define POLY3(x, x2, c2, c1, c0) MLA(x2, C2V(c2), MLA(x, C2V(c1), C2V(c0)))

6.  #define POLY4(x, x2, c3, c2, c1, c0) MLA(x2, MLA(x, C2V(c3), C2V(c2)), MLA(x, C2V(c1), C2V(c0)))

7.  #define POLY5(x, x2, x4, c4, c3, c2, c1, c0) MLA(x4, C2V(c4), POLY4(x, x2, c3, c2, c1, c0))

8.  #define POLY6(x, x2, x4, c5, c4, c3, c2, c1, c0) MLA(x4, POLY2(x, c5, c4), POLY4(x, x2, c3, c2, c1, c0))

9.  #define POLY7(x, x2, x4, c6, c5, c4, c3, c2, c1, c0) MLA(x4, POLY3(x, x2, c6, c5, c4), POLY4(x, x2, c3, c2, c1, c0))

10. #define POLY8(x, x2, x4, c7, c6, c5, c4, c3, c2, c1, c0) MLA(x4, POLY4(x, x2, c7, c6, c5, c4), POLY4(x, x2, c3, c2, c1, c0))

---

The above polynomial calculation scheme is an evaluation scheme for a polynomial of order 7, i.e., $a_7 x^7 + a_6 x^6 + \ldots + a_1 x + a_0$, simd_vmad () is the multiplication and addition instruction, POLY8 represents the arithmetic function of a polynomial of order 7, c0 represents a0, c1 represents a1, and so on, and so on.

## 5. Results and Discussion

### 5.1. Experimental Setup

All the experiments in this study were performed on the compute nodes of SW26010-Pro [10]. The experimental configurations are listed in Table 4.

**Table 4.** Experimental setup.

| Environment | Item | Item Details |
|---|---|---|
| Hardware | Processor | SW26010-Pro |
| | Master Core freq. | 2.1 GHz |
| | Slave Core freq. | 2.1 GHz |
| | LDM cache size | 256 KB |
| Software | Compiler | SWGCC |
| | Python version | 3.6.8 |
| | PyTorch version | 1.5.0 |
| | TensorFlow version | 1.15.0 |

*5.2. Accuracy Tests of Activation Functions*

In this study, based on the high precision computing multiple precision floating-point reliable (MPFR) library [41], the maximum errors of the three precision formats were tested with 16 functions. The errors are reported in ulps, and the accuracy test results are listed in Table 5.

**Table 5.** Test environment.

| Function Name | Maximum Error (Unit: Ulp) | | |
|---|---|---|---|
| | Single-Precision Master Core | Single-Precision Slave Core | Half-Precision = Slave Core |
| Hardshrink | 0.00 | 0.00 | 0.00 |
| ReLU6 | 0.00 | 0.00 | 0.00 |
| ReLU | 0.00 | 0.00 | 0.00 |
| ThresholdedReLU | 0.00 | 0.00 | 0.00 |
| Softshrink | 0.00 | 0.00 | 0.00 |
| PreLU | 0.00 | 0.00 | 0.00 |
| Softsign | 0.00 | 0.00 | 0.00 |
| **ELU** | 0.83 | 0.00 | 0.00 |
| Sigmoid | 0.00 | 0.50 | 0.50 |
| LeakyReLU | 0.00 | 0.00 | 0.00 |
| Swish | 0.00 | 0.00 | 0.00 |
| **LogSigmoid** | 3.00 | 3.00 | 0.00 |
| **SeLU** | 1.57 | 1.57 | 0.00 |
| Softplus | 0.00 | 0.00 | 0.00 |
| Hardtanh | 0.00 | 0.00 | 0.00 |
| HardSigmoid | 0.00 | 0.00 | 0.00 |

The results indicate that the maximum error for the majority of functions was less than 0.5 ulps (0 ulp was the most common result), which was considered as correct rounding achieved. ELU, LogSigmoid, and SeLU functions were the exceptions. To fully investigate these three functions, distribution of the errors across different intervals was tested. The results are listed in Table 6.

**Table 6.** Percentage of errors in each interval for functions with errors higher than 0.5 ulp.

| Version of the Function | Max. Error | 0 Ulp | [0, 0.5] Ulp | [0.5, 1] Ulp | [1, 2] Ulp | [2, 3] Ulp |
|---|---|---|---|---|---|---|
| ELU_SingleSlave | 0.83 | 96.51% | 3.49% | 0% | 0% | 0% |
| LogSigmoid_SingleMaster | 3.00 | 94.82% | 3.35% | 0% | 0% | 1.83% |
| LogSigmoid_SingleSlave | 3.00 | 97.63% | 2.32% | 0% | 0% | 0.05% |
| SeLU_SingleMaster | 1.57 | 92.61% | 6.58% | 0.81% | 0% | 0% |
| SeLU_SingleSlave | 1.57 | 93.22% | 6.03% | 0.75% | 0% | 0% |

It can be observed that more than 98% of the errors for each version of the three tested functions were smaller than 0.5 ulp even though some greater errors were also found. Therefore, it can be considered that these functions also meet the computing accuracy requirements of AI applications.

### 5.3. Performance Test of Activation Functions

Main text paragraph (M_Text). Performance tests of the swAFL library implemented in this study were benchmarked against the PyTorch and TensorFlow libraries ported onto the Sunway platform. The time required by each library to process the same test set was measured, and the speedup ratio was designated as the comparison criterion, which was calculated as speedup = SW_time/swAFL_time, where $SW\_time$ represents the time consumed by SW_PyTorch or SW_TensorFlow, and $swAFL\_time$ represents the time consumed by the functions in swAFL. The result plots are shown in Figure 4.

Figure 4a shows the benchmarking results obtained of the activation functions in the swAFL library against those in SW_PyTorch and SW_TensorFlow, through single-precision floating-point calculations on the master core. From the plot, negative accelerations of swAFL against the SW_PyTorch solution were found in the ELU and Softplus functions; whereas swAFL outperformed SW_TensorFlow using the same functions. Considerable accelerations of swAFL against either SW_PyTorch or SW_TensorFlow were observed with other functions besides ELU and Softplus. To summarize, the speedup ratios of swAFL against SW_PyTorch and SW_TensorFlow were 6.4 and 14.2, respectively. It is worth noting that functions with only one speedup ratio occurred because of the lack of such functions in the benchmark library.

Figure 4b shows the benchmarking results of the same libraries with single-precision floating-point calculations on the slave cores. It can be observed that negative acceleration only occurred with the Softplus function, in the scenario of swAFL against SW_PyTorch; whereas the speedup ratio was positive against SW_TensorFlow. For the ELU function, unlike the result on the master core, over 35% acceleration was observed on the slave cores of swAFL against SW_PyTorch. Considerable accelerations were also noted when running the remaining functions. The speedup ratio using this configuration was 15.2 against SW_PyTorch, and 32.1 against SW_TensorFlow.

Figure 4c shows the benchmarking results of the same libraries with half-precision floating-point calculations on the slave cores, which are similar to the results shown in Figure 4b wherein negative acceleration only occurred in the case of swAFL against SW_PyTorch. Excluding that, more significant improvements were observed, with the speedup ratio of swAFL against SW_PyTorch and SW_TensorFlow being 37.6 and 24, respectively.

Overall, compared with the SW_TensorFlow library, the average performance was improved by 23 times by swAFL; when compared with the SW_PyTorch library, the average performance was improved by 19.5 times though some functions showed negative acceleration. These results demonstrate the high efficiency of the activation function library swAFL.

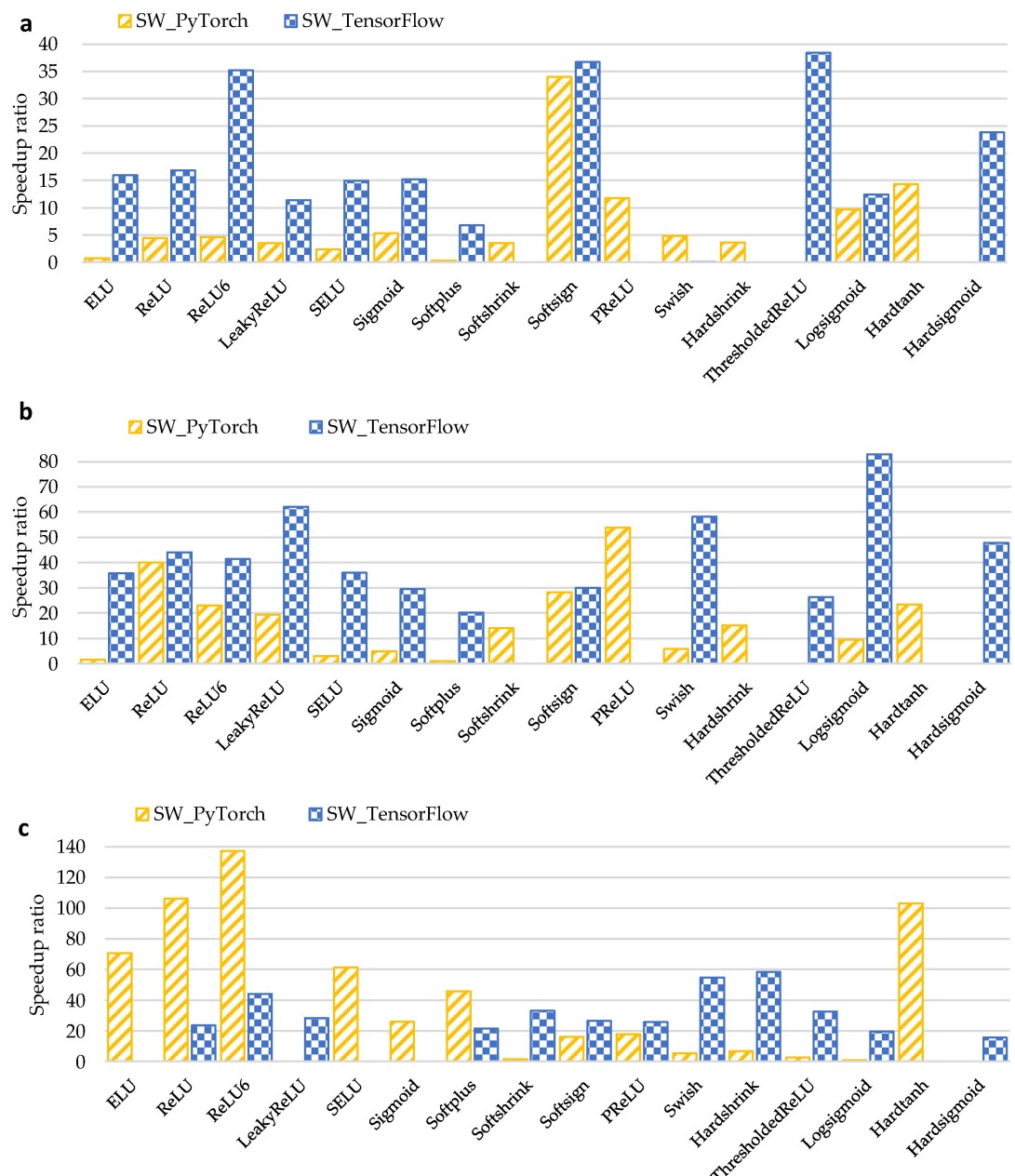

**Figure 4.** Performance benchmarking of the activation functions in the swAFL library. (**a**) Single-precision on the master core, (**b**) Single-precision on the slave core, and (**c**) Half-precision on the slave core.

## 6. Conclusions

In this paper, we designed and implemented a new library called swAFL for the Sunway architecture, fully combining the SIMD extensions and fast multiplication and addition instructions supported by the platform. We also proposed new function design algorithms for different computational accuracies. The experimental results show that the 16 activation functions included in swAFL can meet the user's precision requirements. Compared with the SW_TensorFlow and PyTorch on the same platform, the average performance of swAFL is improved by 23 times and 19.5 times respectively. In addition, the proposed "interval conversion-polynomial approximation" algorithm for single-precision floating-point numbers and the "interval division-polynomial approximation-look-up table" algorithm for half-precision floating-point numbers are also applicable to other architectures.

However, there are still some areas for improvement in the current work. First, the method in this paper has certain limitations. It is universal to this type of activation function of Sigmoid and ReLU, but it is not applicable to other types of activation functions such as VAF and APL. The next step will be to analyze and implement other types of activation functions to further improve the software ecology of the Sunway platform; The second is that when implementing the activation function related to the transcendental function, the half-precision version involves single-precision operations to control the error, wherein the type-conversion step has a significant impact on the performance. Therefore, error control is one of the major tasks to be researched in future work; Third, compared with the SW_TensorFlow library ported onto the platform, the performance of two functions, ELU and Softplus, was relatively poorer than others. Further optimization of these functions is also required.

**Author Contributions:** All authors contributed to the study conception and design. Material preparation, data collection and analysis were performed by F.L., M.H. and P.W. The first draft of the manuscript was written by J.X. and all authors. All authors have read and agreed to the published version of the manuscript.

**Funding:** This research received no external funding.

**Institutional Review Board Statement:** Not applicable.

**Informed Consent Statement:** Informed consent was obtained from all subjects involved in the study.

**Data Availability Statement:** The datasets generated during and/or analysed during the current study are available from the corresponding author on reasonable request.

**Conflicts of Interest:** The authors declare no conflict of interest.

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
