# Peer review of "swAFL: A Library of High-Performance Activation Function for the Sunway Architecture"

_electronics, doi:10.3390/electronics11193141_

Round 1

Reviewer 1 Report

The study designed and implemented a new activation function library called swAFL for the Sunway platform.
The study, in my opionion, could be improved. For instance:

- section 2.3:I don't agree with the proposed activation functions categorisation where only the sigmoid and rectified linear unit are taken into account, for example another important category of activation functions, the trainable activation functions, are not mentioned (see for example [1] for further details). On this topic, it should be interested to know if famous trainable activation functions, such as PELU [2], VAF [3], APL [4], should be made using the proposing framework, or it can be useful only with traditional fixed-shape activation functions.

- introduction:"it outperformed the PyTorch library on the same platform by 6.4, 15.2, and 37.6 times, and the TensorFlow library
by 14.2, 32.1, and 24 times, for single-precision operations on the master core, and half- and single-precision operations on the slave cores, respectively the three types of precision configurations, respectively." this sentence should  be better built

- section 3.1: what is Sollya?? the citation must be made together with the first mention. Furthermore, a brief description should be appreciated.
- "using the high-precision library to calculate the true value of the func tion store the true value into the table calculate the offset to obtain the data in the table." this sentence should be better expressed

- Figure 3 should be better described in its caption.

minor:
row 153: "alogorithm" -> algorithm?

[1] Apicella, Andrea and Donnarumma, Francesco and Isgrò, Francesco and Prevete, Roberto - A survey on modern trainable activation functions, Neural Networks, 2021.
[2] Trottier, L., Gigu, P., Chaib-draa, B., et al. (2017). Parametric exponential linear
unit for deep convolutional neural networks. In Machine learning and ap- plications (ICMLA), 2017 16th IEEE international conference on (pp. 207–214). IEEE.
[3] Apicella, A., Isgrò, F., & Prevete, R. (2019). A simple and efficient architecture for trainable activation functions. Neurocomputing, 370, 1–15.
[4] Agostinelli, F., Hoffman, M. D., Sadowski, P. J., & Baldi, P. (2015). Learning
activation functions to improve deep neural networks. In 3rd international conference on learning representations, ICLR 2015

Reviewer 2 Report

This paper discussed Sunway platform and proposed a design algorithm for different computing accuracies by utilizing the SIMD extensions and FMA operations at the platform level.  Observations are list in the following points and need to fix:

1. The article is well written and well organized but it has many English language issues need to fixed recommended for complete language correction.
2. Some of what is described in this work is available in the literature in particular this article and need to clarify the novel contributions. What is the use of above study and what are major merits and demerits should be analysis.
3. Advised to do comprehensive literature review as many similar works exist in this area.
4. Novelty in model development or development new model or approach is missing. The complete theory part not gives a full impact of readers.
5. Figures or content from other author should be properly cited.
6. Flowchart should be properly draw.
7. The author needs to add a paragraph to describe the
Polynomial generation  feature that used in this research
8. The figure need to be more clear such Figure 3
9. More related reference work should have been taken up.
10. Analysis of the work is weak.
11. Many typo errors in subsections need complete rework.
12. Quality of figures needs to improve.
13. Please check your whole reference list carefully to ensure ALL citations are directly relevant to the specific topic of the manuscript.

14. What is the use of  interval partitioning algorithm , How it is used in your proposed study.

Round 2

Reviewer 1 Report

1) the author states:

 "Due to the variety of activation functions at present, the classification of activation functions in this article is based on the following two points:

1. The nature of the function itself

2. The connection between the implementation algorithm of the function and the algorithm used in this paper to implement the swAFL activation function library function"

I understand the point 2, but not the point 1. What is "the nature of the function itself"?

- "Our classification is more applicable to the swAFL activation function library developed in the article, which is appropriate to the nature of the function and the method of implementation."

if I understand correctly, this is an ad-hoc classification for the proposed framework. However, the existence of other activation functions categorisations in literature should be at least mentioned.

- "However, this algorithm also has certain limitations, such as it is not applicable to the VAF and APL activation functions you mentioned. "

these and other ascertained limitations of the proposed work should be added to the manuscript, or in a proper section or in the conclusions.

After these corrections, in my opinion the paper can be accepted.

Reviewer 2 Report

Based on the revised paper verification, Author has made all the corrections.   Recommended for further publication process.